# Association between Carotid Artery Calcification and Periodontal Disease Progression in Japanese Men and Women: A Cross-Sectional Study

**DOI:** 10.3390/jcm9103365

**Published:** 2020-10-20

**Authors:** Nanae Dewake, Yasuaki Ishioka, Keiichi Uchida, Akira Taguchi, Yukihito Higashi, Akihiro Yoshida, Nobuo Yoshinari

**Affiliations:** 1Department of Operative Dentistry, Endodontology and Periodontology, School of Dentistry, Matsumoto Dental University, Shiojiri 399-0781, Japan; yasuaki.ishioka@mdu.ac.jp (Y.I.); nobuo.yoshinari@mdu.ac.jp (N.Y.); 2Department of Oral Sciences, Matsumoto Dental University Hospital, Shiojiri 399-0781, Japan; keiichi.uchida@mdu.ac.jp; 3Department of Oral and Maxillofacial Radiology, School of Dentistry, Matsumoto Dental University, Shiojiri 399-0781, Japan; akira.taguchi@mdu.ac.jp; 4Department of Cardiovascular Regeneration and Medicine, Research Institute for Radiation Biology and Medicine, Hiroshima University, Hiroshima 734-0037, Japan; yhigashi@hiroshima-u.ac.jp; 5Department Oral Health Promotion, Graduate School of Oral Medicine, Matsumoto Dental University, Shiojiri 399-0781, Japan; akihiro.yoshida@mdu.ac.jp; 6Department of Oral Microbiology, School of Dentistry, Matsumoto Dental University, Shiojiri 399-0781, Japan

**Keywords:** carotid artery calcification, periodontal disease, alveolar bone loss, computed tomography, panoramic radiographs

## Abstract

Objective: To evaluate the association between alveolar bone loss (ABL) detected on panoramic radiographs and carotid artery calcification (CAC) detected on computed tomography (CT). Methods: The study subjects included 295 patients (mean age ± SD: 64.6 ± 11.8 years) who visited the Matsumoto Dental University Hospital. The rate of ABL and the number of present teeth were measured on panoramic radiographs. Univariate analyses with *t*-tests and chi-squared tests were performed to evaluate the differences in age, gender, history of diseases, number of present teeth, and the ABL between subjects, with and without CAC. Moreover, multivariate logistic regression analysis, with forward selection and receiver operating characteristic curve (ROC) analysis, was performed. Results: The number of subjects without and with CAC was 174 and 121, respectively. Univariate analyses revealed that CAC was significantly associated with age, hypertension, osteoporosis, number of present teeth, and ABL. Multivariate logistic regression analysis adjusted for covariates revealed that the presence of CAC was significantly associated with ABL (OR = 1.233, 95% CI = 1.167–1.303). In the ROC analysis for predicting the presence of CAC, the the area under the ROC curve was the highest at 0.932 (95% CI = 0.904–0.960) for ABL, which was significant. Conclusions: Our results suggest that the measurement of ABL on panoramic radiographs may be an effective approach to identifying patients with an increased risk of CAC.

## 1. Introduction

Japan is a super-aged society where 23.2% of the people die from heart and cerebrovascular diseases caused by arteriosclerosis [1]. According to the results of a patient survey conducted by the Ministry of Health, Labor, and Welfare in 2017, arteriosclerotic diseases have been recorded in people aged 30–90 years [2]. It is, therefore, important to determine asymptomatic arteriosclerosis as soon as possible for proactive prevention of vascular diseases in an effort to extend the life span of Japanese individuals.

Periodontitis, which develops principally from the age of 30 years, is a chronic oral inflammatory disease initiated by the accumulation of a bacterial biofilm on tooth surfaces and perpetuated by dysregulated local and systemic inflammatory immune responses [3]. Along with dental caries, different forms of periodontitis represent the two major global oral health burdens on our society [4]. In Japan, the percent of moderate periodontitis in individuals aged ≥65 years with a periodontal pocket depth (PD) of >4 mm was 57.5% in 2016. The presence of periodontitis is directly correlated with the number of present teeth [5]. In particular, periodontitis shares several risk factors with other noncommunicable diseases, including cardiovascular diseases.

The association between periodontal disease and atherosclerosis is well-known in Western countries. However, the epidemiological evidence linking these two phenomena is relatively poor in Japan. Several past studies have suggested an association between periodontal diseases and markers of subclinical atherosclerosis that are used to assess morphological abnormalities such as carotid intimae media thickness (c-IMT) and carotid plaque, as well as the functional abnormalities such as pulse-wave velocity (PWV) and flow-mediated vasodilation (FMD) of the brachial artery induced by reactive hyperemia [6,7,8,9,10,11,12]. Orlandi et al. conducted a systematic review and meta-analysis and found that the diagnosis of periodontal disease was associated with a mean increase in c-IMT of 0.08 mm (95% CI = 0.07–0.09) and a mean difference in FMD of 5.1% in comparison with those in the corresponding controls (95% CI = 2.08–8.11) [13]. They also described a beneficial effect of periodontal treatment on FMD, indicating an improvement in the endothelial function [13].

In vascular calcification, which is the final step in the development of atherosclerosis, the deposition of hydroxyapatite mineral in the arterial wall has been associated with an increased risk of heart disease, stroke, and atherosclerotic plaque rupture [14]. However, there is only limited information available on the association between periodontal diseases and carotid artery calcification (CAC). In fact, there are several imaging diagnostic tools available for assessing CAC. Of these, CAC can be easily and accurately detected by computed tomography (CT) [15]. On the other hand, periodontal disease progression can be assessed as an alveolar bone loss (ABL) on panoramic radiographs. Therefore, the purpose of the present study is to evaluate the association between the rate of ABL measured on panoramic radiographs and CAC detected by CT.

## 2. Methods

### 2.1. Patient Setting

A total of 295 patients (174 men and 121 women) who underwent both panoramic radiographs and CT for the examination of their lesions (such as benign tumors and cystic lesions) and implant placement and who visited the Matsumoto Dental University Hospital between 2014 and 2018 were enrolled in this study. Patients with lesions and alveolar bone destruction were excluded from this study. Moreover, patients who had undergone radiation therapy for their head and neck regions were also excluded. The mean age (SD) of the subjects was 64.6 (11.8) years. The information on age, gender, and medical history of the subjects was obtained from their respective medical records. We also extracted data on the potential risk factors for arteriosclerosis, namely, hypertension, hyperlipidemia, diabetes mellitus, and osteoporosis, from the records.

Since this study was a retrospective cross-sectional study, written informed consents were not obtained from all subjects at the time of taking both panoramic radiographs and CT. Instead, the results of this study were provided to all subjects in accordance with the ethical guidelines of the Ministry of Health, Labor, and Welfare and the Ministry of Education, Culture, Sports, Science, and Technology. The institutional review board for clinical research at Matsumoto Dental University reviewed and approved this study protocol (no. 152).

### 2.2. Assessment of Alveolar Bone Resorption Using Panoramic Radiographs

Panoramic radiographs of all subjects were obtained using the digital AZ3000 device (Asahi X-Ren Kogyo, Kyoto, Japan). The received ray system was a computed radiography system (Konica Minolta, Tokyo, Japan). A periodontist with 6 years of experience examined the number of present teeth and the rate of ABL. The supernumerary tooth and the third molar were excluded from the number of present teeth. The rate of ABL was measured on panoramic radiographs [16]. The distances between the cement–enamel junction (CEJ) and the alveolar crest (AC) and that between the CEJ and the root apex were measured at two sites (i.e., mesial and distal) of each of the present teeth. The apex was defined as the most apically located point of the root. In teeth restored with fillings or crowns, the most apical limit of the restoration was considered to be equivalent to the CEJ. Finally, ABL was calculated as the CEJ–AC/CEJ–apex [17]. The measurements were made for all of the present teeth, excluding implants, supernumerary teeth, and the third molars. The residual roots, without a cap for overdenture, were excluded. Teeth with caries or periapical lesions were not excluded. In the PROKRANK study, the radiographic periodontal condition based on the rate of ABL was classified into three groups: healthy (≥80% remaining bone), mild to moderate (66–79% remaining bone), and severe (<66% remaining bone) bone loss groups [18].

### 2.3. Detection of CAC on CT

CAC was detected on the axial CT images captured by using the multislice CT system Activation 16 (Canon Medical Systems, Tochigi, Japan) at the Matsumoto Dental University Hospital. The presence of CAC near the bifurcation of the common carotid artery (about 2-cm upper and lower) was evaluated independently by two oral and maxillofacial diplomate radiologists (one with >20 years of experience and another with 30 years of experience; Figure 1). In case of different outcomes (such as ectopic calcifications) between the two oral and maxillofacial radiologists, a consensus was reached via discussion.

### 2.4. Statistical Analysis

The patients’ age was divided into 5 groups (30–49, 50–59, 60–69, 70–79, and 80–99 years). The number of present teeth was divided into 3 groups (1–9, 10–19, and ≥20). The rate of ABL (%) was divided into 3 groups (≤20%, 20–34%, and >34%) based on the radiographic periodontal classification [18]. Initially, univariate analyses with *t*-tests and chi-squared tests were used to evaluate the differences in age, gender (binary), history of diseases related to atherosclerosis, number of teeth present, and the rate of ABL (%) between the subjects, without and with CAC. Secondly, multivariate logistic regression analysis, with forward selection adjusted for age, gender (binary), and all other variables significant at *p* < 0.20 in the univariate analyses, were tested to calculate the adjusted odds ratio (OR) and 95% confidence interval (CI) of having CAC based on the ABL classification system. Furthermore, receiver operating characteristic (ROC) curve analysis was employed to clarify how asymptomatic CAC can be identified by age, presence of hypertension, number of present teeth, and the rate of ABL. According to the method suggested by Swets [19], the area under the ROC curve (AUROC) was determined as follows: less accurate (0.5 < AUROC < 0.7), moderately accurate (0.7 < AUROC < 0.9), highly accurate (0.9 < AUROC < 1), and perfect tests (AUROC = 1). All comparisons were two-sided and performed at *p* = 0.05 level of significance. Statistical analysis was performed using SPSS ver. 24.0 for Windows (IBM Japan, Tokyo, Japan).

## 3. Results

The characteristics of all participants, according to their age groups, are shown in Table 1. The prevalence of calcification (*p* < 0.001), risk of osteoporosis (*p* = 0.02), number of present teeth (*p* < 0.001), and ABL (*p* < 0.001) were noted to significantly increase with advancing age. The number of subjects without and with CAC was 174 (99 men and 75 women; Group NC) and 121 (68 men and 53 women; group C), respectively. The mean ages (SD) of Group C and Group NC individuals were 72.0 (9.7) and 59.4 (10.3) years, respectively.

Significant differences were noted in age (*p* < 0.001), history of hypertension (*p* < 0.001), osteoporosis (*p* = 0.004), number of present teeth (*p* < 0.001), and ABL (*p* < 0.001) between Groups C and NC (Table 2). The rate of ABL in Group C was significantly greater than that in Group NC.

Multivariate logistic regression analysis, with forward selection adjusted for covariates, revealed that the presence of CAC was significantly associated with age (OR = 1.096, 95% CI = 1.051–1.143, *p* < 0.001), history of hypertension (OR = 3.748, 95% CI = 1.748–8.037, *p* = 0.001), and ABL (OR = 1.233, 95% CI = 1.167–1.303, *p* < 0.001; see Table 3). In the final model adjusted for the age group and the presence of hypertension, the OR of having CAC in subjects with 20% < ABL ≤ 34% and ABL > 34% was 23.676 (95% CI = 9.494–59.035, *p* < 0.001) and 111.848 (95% CI = 31.322–399.398, *p* < 0.001) in comparison with subjects with ABL ≤ 20%, respectively (Table 3).

In the ROC analysis predicting the presence of CAC, the AUROC was 0.932 (95% CI = 0.904–0.960, *p* < 0.001) for ABL, 0.815 (95% CI = 0.767–0.864, *p* < 0.001) for age, 0.685 (95% CI = 0.692–0.806, *p* < 0.001) for presence of hypertension, and 0.749 (95% CI = 0.621–0.748, *p* < 0.001) for the number of present teeth (Table 4, Figure 2).

## 4. Discussion

This is the first study of its kind to investigate the association between the presence of CAC detected on CT and ABL that demonstrates the progression of periodontal diseases in a Japanese population.

Based on the report by Mattila et al. in 1989, no consensus has been attained on the presence of a causal relationship between periodontal diseases and ischemic heart diseases [20]. However, in their 2007 meta-analysis, Bahekar et al. noted that periodontal diseases were associated with an increased incidence of ischemic heart diseases [21]. In addition, in 2012, the American Heart Association submitted a systematic review of about 500 articles published between 1950 and 2011 that reported associations between periodontal diseases and atherosclerotic vascular disease (ASVD) [22]. Although periodontal interventions reduce the incidences of systemic inflammation and endothelial dysfunction in short-term studies, there is presently no evidence of them preventing ASVD or modifying its outcomes [22]. However, several common risk factors have been reported between periodontal diseases and ischemic heart diseases as confounding factors [23,24]. A 2014 meta-analysis reported that a periodontal disease increases the incidence of cerebrovascular disease [25]. In Japan, Taguchi et al. reported a significant association among the number of lacunar infarctions, a type of cerebral infarction, and ABL [26]. However, a cohort study found no such significant association [27].

In a recent study, periodontal bacterial flora was found to be associated with vascular diseases [28]. Advanced periodontitis in patients with ischaemic stroke is associated with a greater neurological deficit on admission [29], whereas epidemiological studies have reported no association between periodontal PD and myocardial infarction, stroke, or heart failure [30]. These confounding reports suggest that evidence linking periodontitis and cerebrovascular disease is inadequate, considering the lack of a consensus on the definition of a periodontal disease as well as on the objectives of clinical parameters employed in epidemiological studies and/or in interventional studies using standardized treatment protocols. On the other hand, periodontal disease affects short-term systemic inflammatory conditions and vascular endothelial cell functions. Therefore, long-term observational studies are required to clarify this point.

Periodontal diseases are associated with FMD and IMT (noninvasive assessments of vascular functions). The FMD refers to the rate of change in the blood vessel diameter; if the denominator vessel diameter is large, the FMD is relatively low even when the functions are normal. In addition, the image quality is poor in elderly or obese patients, as they tend to have larger upper arm diameters [31]. The diagnoses of early-stage atherosclerosis are usually performed via c-IMT with the detection of carotid artery echoes. Some past reviews have reported no significant association between c-IMT-indicated progression and increased risk for cardiovascular events. CT is slightly invasive, but it reliably detects calcification, which is the final stage of atherosclerosis, and CAC is associated with cardiovascular diseases [32,33,34,35]. Hence, CAC screening is considered more useful than the applications of FMD and c-IMT.

Since Friedlander et al. reported that CAC can be identified in panoramic radiographs [36], several researchers have evaluated the CAC-associated risks for cardiovascular lesions [32,33,34,35]. Accordingly, several positive associations have been reported, including that the CAC status is an effective indicator for cardiovascular lesions [37] as it can predict carotid stenosis [32], which is associated with peripheral arterial diseases (in Koreans aged ≥50 years) [33]. Peripheral arterial diseases are present in 84% of the patients with carotid artery stenosis [34] and are associated with carotid atherosclerosis [35]. Thus, a relationship exists between the assessment of CACs on digital panoramic radiographs and periodontitis [28,38,39]. However, Thanakun et al. reported that ABL was not associated with CAC [40]. In this study, we diagnosed CAC using CT because CT could be used to diagnose CAC with higher probability than panoramic radiography [41]. Moreover, we noted a significant association between the number of present teeth and CAC. However, ROC analyses revealed that ABL was a better screening factor than the number of present teeth.

We identified some limitations in this study. First, all subjects visited the Matsumoto Dental University Hospital and, therefore, probably lived in a specific region of Japan, which makes them nonrepresentative of the entire Japanese population. This factor may have introduced selection bias. Second, although we extracted the age and the status of hypertension, dyslipidemia, diabetes, osteoporosis, and cancer of the subjects from their medical records, we lacked data on their smoking status and the levels of C-reactive protein, total cholesterol, and high-density lipoprotein, which are all associated with CAC. In the Framingham heart study, 5573 first- and second-generation patients lacked cardiovascular diseases, high blood pressure, total cholesterol level, smoking, glucose-tolerance, and left ventricular hypertrophy, which are all risk factors for CVD progression [42]. Therefore, these factors may have also affected the CAC progression. We intend to explore this topic in the future. Third, we evaluated the periodontal disease status using a single method. Usually, periodontal PD, clinical attachment level (CAL), and bleeding on probing (BOP) are some of the factors used to diagnose periodontal diseases. However, as several of our patients underwent oral surgery, we lacked data on these points. We instead explored whether the CAC status was evident from the obtained panoramic images. We noted that ABL was more convenient to assess than PD, CAL, and BOP. Digital panoramic radiograph grades were also in good agreement with other radiographic, periodontal-disease-grading methods [43,44] and compatible with other measurement methods of periodontal diseases and other diagnostic methods [45,46,47]. We plan to develop a tool that measures the ABL rate in panoramic radiographs. Fourth, as this was a cross-sectional study, we could not clarify the presence of any causal relationship between CAC and ABL. This point requires further longitudinal study.

The strength of our study is that we enrolled 295 patients of different ages, unlike in previous related studies. In the ROC analyses, the AUROCs for ABL, age, hypertension, and the number of teeth were 0.932, 0.815, 0.685, and 0.749, respectively, suggesting that ABL and age may be used to screen for individuals with CAC. To the best of our knowledge, only a few epidemiological surveys have been conducted to evaluate the association between CAC and ABL in Japan.

## 5. Conclusions

The measurement of alveolar bone resorption in panoramic radiographs may effectively identify patients at increased risk for CAC. Further cross-sectional and longitudinal studies on a large scale, involving the exploration of important covariates such as smoking, are warranted in the future.

## Figures and Tables

**Figure 1 jcm-09-03365-f001:**
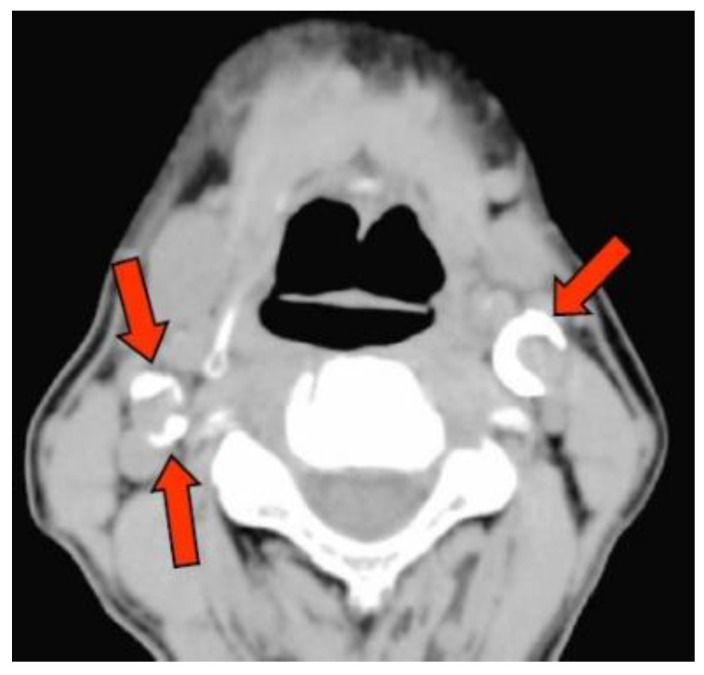
Observation of the carotid artery calcification. Arrow tip indicates the calcification.

**Figure 2 jcm-09-03365-f002:**
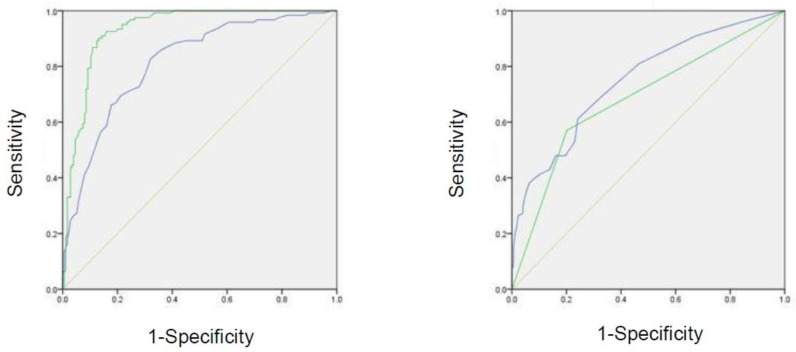
ROC curve of effective factors for the screening of carotid artery calcification. Green line in the left panel represents the rate of ABL. Blue line in the left panel represents the age. Green line in the right panel represents hypertension. Blue line in the right panel represents the number of present teeth.

**Table 1 jcm-09-03365-t001:** The characteristics of all participants by age group.

	Age							
	30–39 y (*n* = 2)	40–49 y (*n* = 32)	50–59 y (*n* = 65)	60–69 y (*n* = 85)	70–79 y (*n* = 82)	80–89 y (*n* = 24)	90–95 y (*n* = 5)	*p*-Value
Male	1 (50.0)	16 (50.0)	40 (61.5)	54 (63.5)	42 (51.2)	11 (45.8)	3 (60.0)	0.40
Female	1 (50.0)	6 (50.0)	25 (38.5)	31 (36.5)	40 (48.8)	13 (54.2)	2 (40.0)	
Calcification	0 (0.0)	2 (6.3)	11 (16.9)	28 (32.9)	55 (67.1)	20 (83.3)	5 (100.0)	<0.001
Not Calcification	2 (100.0)	30 (93.8)	54 (83.1)	57 (67.1)	27 (32.9)	4 (16.7)	0 (0.0)	
Hypertension Yes	0 (0.0)	3 (9.4)	14 (21.5)	31 (36.5)	37 (45.1)	15 (62.5)	4 (80.0)	0.15
No	2 (100.0)	29 (90.6)	51 (78.5)	54 (63.5)	45 (54.9)	9 (37.5)	1 (20.0)	
Dyslipidemia Yes	0 (0.0)	1 (3.1)	10 (15.4)	12 (14.1)	10 (12.2)	2 (8.3)	1 (20.0)	0.68
No	2 (100.0)	31 (96.9)	55 (84.6)	73 (85.9)	72 (87.8)	22 (91.7)	4 (80.0)	
Diabetes mellitus Yes	0 (0.0)	1 (3.1)	6 (9.2)	14 (16.5)	13 (15.9)	2 (8.3)	2 (40.0)	0.58
No	2 (100.0)	31 (96.9)	59 (90.8)	71 (83.5)	69 (84.1)	22 (91.7)	3 (60.0)	
Osteoporosis Yes	0 (0.0)	0 (0.0)	2 (3.1)	1 (16.5)	13 (15.9)	2 (8.3)	1 (20.0)	0.02
No	2 (100.0)	32 (100.0)	63 (96.9)	84 (83.5)	69 (84.1)	22 (91.7)	4 (80.0)	
Cancer Yes	0 (0.0)	1 (3.1)	3 (9.2)	14 (1.2)	7 (8.5)	5 (20.8)	0 (0.0)	0.38
No	2 (100.0)	31 (96.9)	62 (90.8)	71 (98.8)	75 (91.5)	19 (79.2)	5 (100.0)	
Number of present teeth	27.5 ± 0.7	25.0 ± 4.6	23.9 ± 4.3	21.9 ± 5.7	17.7 ± 7.8	14.5 ± 6.5	11.4 ± 8.8	<0.001 ^a^
1–9	0 (0.0)	1 (3.1)	1 (1.5)	3 (3.5)	16 (19.5)	7 (29.2)	2 (40.0)	0.002
10–19	0 (0.0)	3 (9.4)	7 (10.8)	20 (23.5)	22 (26.8)	9 (37.5)	1 (20.0)	
≥20	2 (100.0)	28 (87.5)	57 (87.7)	62 (72.9)	44 (53.7)	8 (33.3)	2 (40.0)	
ABL	19.8 ± 12.4	17.2 ± 7.7	19.4 ± 9.7	21.1 ± 9.3	29.5 ± 12.3	29.6 ± 9.2	35.7 ± 7.4	<0.001 ^a^
≤20%	1 (50.0)	26 (81.3)	44 (67.7)	49 (57.6)	19 (23.2)	5 (20.8)	0 (0.0)	<0.001
>20%, ≤34%	1 (50.0)	5 (15.6)	15 (23.1)	28 (32.9)	38 (46.3)	11 (45.8)	2 (40.0)	
>34%	0 (0.0)	1 (3.1)	6 (9.2)	8 (9.4)	25 (30.5)	8 (33.3)	3 (60.0)	

^a^*t*-test: mean ± standard deviation; chi-square test: *n* (%); ABL: the rate of alveolar bone loss.

**Table 2 jcm-09-03365-t002:** The relationship between carotid artery calcification and other variables.

	Group C	Group NC	*p*-Value
	(*n* = 121) Male: 68, Female: 53	(*n* = 174) Male: 99, Female: 75	
Age (Year)	72.0 ± 9.7	59.4 ± 10.3	<0.001 ^a^
30–49 y	2 (5.9)	32 (94.1)	<0.001
50–59 y	11 (16.9)	54 (83.1)	
60–69 y	28 (32.9)	57 (67.1)	
70–79 y	55 (67.1)	27 (32.9)	
80–95 y	25 (86.2)	4 (13.8)	
Male	68 (56.2)	99 (56.9)	0.91
Hypertension	69 (57.0)	35 (20.1)	<0.001
Dyslipidemia	14 (11.6)	22 (12.6)	0.78
Osteoporosis	12 (9.9)	4 (2.3)	0.004
Diabetes Mellitus	19 (15.7)	19 (10.9)	0.23
Cancer	13 (10.7)	19 (10.9)	0.96
Number of Present Teeth	17.1 ± 7.9	23.3 ± 4.8	<0.001 ^a^
1–9	27 (90.0)	3 (10.0)	<0.001
10–19	31 (50.0)	31 (50.0)	
≥20	63 (31.0)	140 (69.0)	
ABL (%)	32.7 ± 9.7	17.2 ± 7.0	<0.001 ^a^
≤20%	8 (5.6)	136 (94.4)	<0.001
>20%, ≤34%	67 (67.0)	33 (33.0)	
>34%	46 (90.2)	5 (9.8)	

^a^*t*-test: mean ± standard deviation; chi-square test: *n* (%); ABL: the rate of alveolar bone loss.

**Table 3 jcm-09-03365-t003:** Factors associated with carotid artery calcification and the rate of alveolar bone loss evaluated by multivariate logistic regression analysis using forward selection.

		Partial Regression Coefficient	Standard Error	Odds Ratio (95% CI)	*p*-Value
Step 1	ABL	0.231	0.026	1.260 (1.197–1.325)	<0.001
	constant	−5.744	0.614	0.003	<0.001
Step 2	Age	0.099	0.020	1.105 (1.062–1.149)	<0.001
	ABL	0.214	0.028	1.239 (1.173–1.308)	<0.001
	Constant	−11.916	1.572	0.000	<0.001
Step 3	Age	0.092	0.021	1.096 (1.051–1.143)	<0.001
	Hypertension	1.321	0.389	3.748 (1.748–8.037)	0.001
	ABL	0.210	0.028	1.233 (1.167–1.303)	<0.001
	Constant	−11.883	1.658	0.000	<0.001
Step 1	ABL ≤20%			1.000	<0.001
	>20%, ≤34%	3.541	0.421	34.515 (15.112–78.833)	<0.001
	>34%	5.052	0.595	156.400 (48.722–502.053)	<0.001
	Constant	−5.744	0.614	0.003	<0.001
Step 2	Age 30–49 y			1.000	<0.001
	50–59 y	0.872	0.924	2.392 (0.391–14.617)	0.345
	60–69 y	1.937	0.883	6.940 (1.128–39.206)	0.028
	70–79 y	2.598	0.882	13.437 (2.387–75.654)	0.003
	80–95 y	4.188	1.102	65.902 (7.606–571.037)	<0.001
	ABL ≤20%			1.000	<0.001
	>20%, ≤34%	3.367	0.460	28.988 (11.759–71.456)	<0.001
	>34%	4.747	0.643	115.220 (32.686–406.154)	<0.001
	Constant	−4.705	0.905	0.009	<0.001
Step 3	Age 30–49 y			1.000	<0.001
	50–59 y	0.534	0.934	1.705 (0.273–10.638)	0.568
	60–69 y	1.498	0.893	4.471 (0.776–25.758)	0.094
	70–79 y	2.194	0.889	8.974 (1.571–51.274)	0.014
	80–95 y	3.747	1.130	42.410 (4.626–388.797)	0.001
	Hypertension	1.026	0.399	2.790 (1.275–6.104)	0.001
	ABL ≤20%			1.000	<0.001
	>20%, ≤34%	3.164	0.466	23.676 (9.494–59.035)	<0.001
	>34%	4.717	0.649	111.848 (31.322–399.398)	<0.001
	Constant	−11.883	1.658	0.000	<0.001

CI: confidence interval; ABL: the rate of alveolar bone loss.

**Table 4 jcm-09-03365-t004:** Effective factors for the screening of carotid artery calcification using ROC analysis.

Covariance	AUROC	Standard Error	*p*-Value (95% CI)
ABL	0.932	0.014	<0.001 (0.904–0.960)
Age	0.815	0.025	<0.001 (0.767–0.864)
Hypertension	0.685	0.032	<0.001 (0.621–0.748)
Number of Present Teeth	0.749	0.029	<0.001 (0.692–0.806)

CI: confidence interval; ABL: the rate of alveolar bone loss; AUROC: area under the receiver operating characteristic curve.

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
