# Peer review of "Association between Carotid Artery Calcification and Periodontal Disease Progression in Japanese Men and Women: A Cross-Sectional Study"

_jcm, 2020, doi:10.3390/jcm9103365_

Round 1

Reviewer 1 Report

Dear Authors,

Congratulation on your work with big number of the subjects, and selected topic which is still crucial and influential. Periodontitis is an independent risk factor for atherosclerosis and consequently for cardiac or stroke development. It is very important to try find out simple and useful classification or methods to prevent common systemic diseases.

The methodology is well written and enables its repeatability. The results are well presented on the tables and graphs. Despite the lack of innovation, its results are satisfactory which makes the work publishable. The study is interesting, however, some minor changes need to be addressed, like in all manuscript please correct all extra spaces, especially table (line 187), which cover manuscript text up. It could be worth considering to add citation of the valuable clinical paper reported that periodontitis affects neurological deficit in acute stroke [doi: 10.1016/j.jns.2010.07.012].

Reviewer 2 Report

Review of JCM 951338

The manuscript needs considerable work to correct language misuse and syntax errors. The authors are urged to have a person fluent in written English edit the manuscript before submission of the final version.

The connections between calcification and ABL can be co-incidental, not related. The manuscript must emphasize that this association needs further investigation by longitudinal epidemiologic and interventional trials to prove if the association is related, and if indeed periodontal inflammation contributes to the cause of calcification.

Why would the connection between atherosclerosis and periodontal inflammation not be similar for Japanese people as it has been documented for other ethnic and geographically dissimilar groups? Shouldn’t the basic biology that could account for the connection be shared by all humans?

As this is a cross-sectional study, only prevalence can be estimated. Yet, the authors use terms like rate – for example “Characteristics of all participants according to age group were shown in Table 1. The rate of 134 calcification (p < 0.001), risk of osteoporosis (p = 0.02), number of present teeth (p < 0.001), and ABL (p 135 < 0.001) significantly increased with advancing age.” This is not correct. Rate can only be determined by longitudinal studies. Please correct this throughout the manuscript.

The observation reported here is not novel – please cite previous investigations for ezample those below and others.

Assessment of carotid artery calcifications on digital panoramic radiographs and their relationship with periodontal condition and cardiovascular risk factors.

Bilgin Çetin M, Sezgin Y, Nisancı Yilmaz MN, Köseoğlu Seçgin C.Int Dent J. 2020 Sep 29. doi: 10.1111/idj.12618.

Carotid artery calcification in panoramic radiographs associates with oral infections and mortality. Paju S, Pietiäinen M, Liljestrand JM, Lahdentausta L, Salminen A, Kopra E, Mäntylä P, Buhlin K, Hörkkö S, Sinisalo J, Pussinen PJ.Int Endod J. 2020 Aug 31. doi: 10.1111/iej.13394. 

Evaluation of the relationship between periodontal risk and carotid arterycalcifications on panoramic radiographs.

Kamak G, Yildirim E, Rencber E.Eur J Dent. 2015 Oct-Dec;9(4):483-489.
